# Functionalization of Conductive Polymers through Covalent Postmodification

**DOI:** 10.3390/polym15010205

**Published:** 2022-12-31

**Authors:** Silvestre Bongiovanni Abel, Evelina Frontera, Diego Acevedo, Cesar A. Barbero

**Affiliations:** 1Research Institute for Materials Science and Technology (INTEMA), National University of Mar del Plata (UNMdP)-National Council of Scientific and Technical Research (CONICET), Mar del Plata 7600, Argentina; 2Departamento de Ciencias Aplicadas y Tecnología, Escuela de Ingeniería y Ciencias Ambientales, Universidad Nacional de Villa Mercedes, Villa Mercedes 5730, Argentina; 3Research Institute for Energy Technologies and Advanced Materials (IITEMA), National University of Río Cuarto (UNRC)-National Council of Scientific and Technical Research (CONICET), Río Cuarto 5800, Argentina

**Keywords:** conducting polymers, covalent modifications, solubility, electrophilic reaction, nucleophilic substitution

## Abstract

Organic chemical reactions have been used to functionalize preformed conducting polymers (CPs). The extensive work performed on polyaniline (PANI), polypyrrole (PPy), and polythiophene (PT) is described together with the more limited work on other CPs. Two approaches have been taken for the functionalization: (i) direct reactions on the CP chains and (ii) reaction with substituted CPs bearing reactive groups (e.g., ester). Electrophilic aromatic substitution, S_E_Ar, is directly made on the non-conductive (reduced form) of the CPs. In PANI and PPy, the N-H can be electrophilically substituted. The nitrogen nucleophile could produce nucleophilic substitutions (SN) on alkyl or acyl groups. Another direct reaction is the nucleophilic conjugate addition on the oxidized form of the polymer (PANI, PPy or PT). In the case of PT, the main functionalization method was indirect, and the linking of functional groups via attachment to reactive groups was already present in the monomer. The same is the case for most other conducting polymers, such as poly(fluorene). The target properties which are improved by the functionalization of the different polymers is also discussed.

## 1. Introduction

Conducting polymers (CP) [1] are modern materials that hold the promise of simultaneously possessing properties typical of metals (e.g., high electrical conductivity [2]) and of non-conductive polymers: thermoplasticity [3], chemical stability in air [4], and good mechanical properties [5]. It was quickly discovered that typical polymer and metal properties are mutually exclusive since the bulk conductivity of linear polymers requires orbital overlapping between chains (strong interaction), leading to rigid and fragile materials with high glass transition temperatures (Tg). Accordingly, there are no known applications of CPs as massive electric conductors. However, they have been applied as flexible electrical conductors in microelectronics [6] and are widely applied as “conductive inks” [7] for micro- and nanotechnological applications. Moreover, it has been shown that other properties—electrochromism, redox electroactivity, redox-driven ion exchange, remote heating/blocking of electromagnetic radiation absorption, semiconductivity, interaction with small molecules, charge storage, controlled light emission, etc., are technologically as valuable as conductivity [1]. Indeed, conducting polymers can be used in different applications: electromagnetic shielding [8], electrochromic [9,10], electromagnetic actuators [11], electrode materials of batteries [12,13] or supercapacitors [6], conducting hydrogels [14], electrochemically driven ion exchangers [15], photothermal antitumoral/antibacterial therapy [16,17,18], electrochemical sensors [19], etc. Moreover, being organic materials, they are more likely to be biocompatible and biodegradable than metals or semiconductors since no toxic elements will be produced during degradation. 

Being organic macromolecules, they also hold the promise of being easily interfaced with by biological entities [20,21] since the CPs can, via well-known organic reactions, covalently link biomolecules to the CP backbone. As we will see, the attached groups could add other valuable properties, such as solubility in common solvents (including aqueous solutions), fluorescence emission, additional charge storage, additional electrochromic centers, complexing groups, improved wettability and stability, self-doping, etc. However, the electrical conductivity of functionalized polymers is usually lower than that of the base CP due to several factors: (i) Since charge carriers in p-doped CPs are cation radicals, the attachment of negatively charged groups made it a self-doped polymer (i.e., has fixed counterions linked to the chain which cannot be lost) but pin the charge carriers to fixed locations, reducing their mobility [22]. Any groups which donate or withdraw electrons from an aromatic ring could localize the charge [23]. (ii) Since CPs are made of aromatic rings linked in chains with extended conjugation, the presence of bulky substituents induces steric effects, which diminish the ring-to-ring planarity decreasing extended conjugation and intrachain electronic conductivity [24]. Bulk electronic conductivity not only requires high intrachain conductivity (related to extended conjugation and high charger carrier mobility) but also interchain electron transfer related to π–π overlapping. The presence of any group increases the chain-to-chain distance due to steric effects and decreases interchain conductivity. Therefore, polymer functionalization is likely to decrease bulk polymer conductivity. On the other hand, additional properties (fluorescence, complexation, hydrophobicity, self-doping) could be present in the functionalized polymer. Moreover, significant improvements of intrinsic properties (electroactivity, redox coupled ion exchange, electrochromism) are usually obtained. Therefore, the trade-off of some conductivity loss is accepted. In most technological applications, electrical conductivity is not the key parameter, while other properties and/or improved polymer processability are more relevant. Extended conjugation along the chain also makes the CP rigid, which could be useful in mechanical applications but strongly decreases the conformational entropy in solution, making solubility thermodynamically unfavorable. A simple method of processing CP films involves preparing solutions in specific solvents and then producing film via solvent evaporation (spin coating, drop casting, etc.) [25]. However, unmodified CPs are either insoluble (PT, PPy) or soluble in some special solvents (PANI in uncharged forms (e.g., PANI(EB)), making them difficult to process. Additionally, having stiff chains (due to the extended conjugation) makes them non-plastic, which does not allow the thermomechanical (e.g., extrusion) formation of films. Solutions of CPs can also be used to produce CP nanoparticles by mixing the solution with an antisolvent which is miscible with the CP’s solvent but does not dissolve it [26]. Dissolution of rigid polymers, like CPs, is difficult because there is almost no entropic contribution. Unlike small molecules, the number of particles of a high molecular weight polymer in solution is low and does not contribute to entropy. Unlike flexible polymers (e.g., poly(styrene))—for which free rotation in the C-C bond creates a number of different conformations, contributing to entropy—rigid polymers have few conformations. Therefore, the solubility is determined by enthalpy related to polymer–polymer (endothermic), polymer–solvent (exothermic), and solvent–solvent (endothermic) interactions. When solvent–solvent enthalpy is large (e.g., in water), few polymers are soluble. 

A substituent attached to the polymer chain could increase solubility due to several factors: (i) the steric effect of the substituent decrease chain-to-chain interactions, making it energetically favorable to separate the chains; (ii) the attached functional group (e.g., –R–OH) could have specific interactions with the solvent (e.g., hydrogen bonding) which increase solubility. The attachment of fixed charged groups (e.g. –SO_3_^−^) decreases the interaction between chains due to coulombic repulsion, increasing dissolution. Moreover, the ionic charge has strong ion–dipole interactions with the polar solvent (water), increasing solubility. In aqueous electrolytes, the presence of fixed charges requires compensation by mobile counterions. Soluble ions (e.g., Cl^−^) do not form ion pairs with the charged chains but create an ionic region of solvated ions (double-layer region, rich in counterions but containing coions) whose thickness is inversely related to the ionic force of the solution [27]. The process increases the solubility since the whole double layer is the hydration sphere of the ions. If the charge of the attached group is negative, the group will interact coulombically with the positive charge carriers (in p-doped CPs). This phenomenon, called self-doping, implies that no mobile counterions are present and that the solubility decreases. Elimination of the positive charge carriers (by a reduction in all CPs or deprotonation in the case of PANI and PPy) implies the formation of the ionic double layer and increases solubility. This is the reason that sulfonated polyaniline (SPAN), in its EB state is soluble in ammonia solution. Solvent evaporation and decomposition of the ammonium salt (with loss of NH_3_) produces thin films which are insoluble in acid media [12]. In some CPs (e.g., PANI) the analogy is closer since both CE and PANI are insoluble in water but are strongly wettted by the solvent due to hydrogen bonding. In the case of cellulose, such interaction affects strongly the properties of the fibers (plasticizing effect) [28]. In the case of PANI, the electrical conductivity of films is strongly affected by the humidity level. The analogy extends to the use of salt/solvent (e.g., Li^+^ salt/NMP) to avoid aggregation of cellulose [4] or PANI chains [29,30]. Moreover, the solubility of cellulose in common solvents can be improved through the use of acids or bases [4]. The same is true for PANI, for which solubility has been induced by adding acid [31], and bases [32]. It is known that PANI forms strong hydrogen bonds between neighboring chains [33] and with solvents [34]. PPy should also be able to form hydrogen bonds, similarly to pyrrole [35]. Since thiophene monomer units do not form hydrogen bonds, PT is insoluble. It is well known that substituted celluloses (e.g., methylcellulose) are soluble in water, while the added functional group (–OCH_3_) interacts more weakly with water than the group replaced (–OH) [36]. The role of the added group is mainly to decrease the hydrogen bonding interchain interactions [37]. Since the conductive forms of the CPs are salts, the counterion can be bulky (impeding the chain-to-chain interactions) and/or strongly interactive with the solvent, inducing the so-called counter-ion-induced processability (solubility) [38]. However, reactions with highly reactive reagents (e.g., SO_3_/H_2_SO_4_) could not only produce the desired functionalization but also other reactions and even chain scission. 

A drawback of polymer synthesis shared by functionalization is that macromolecular byproducts cannot be separated, and will be present in the final materials. CPs are usually applied as solid materials (film, pellets, particles, fibers, etc.). Therefore, functionalization could be used to change not only bulk but also surface properties. Since CPs are mostly insoluble (e.g., PPy and PT) or soluble in a few solvents (e.g., PANI), most reactions are carried out heterogeneously. Therefore, the surface can be functionalized without altering the bulk. In that way, the unmodified polymer with good bulk property (e.g., conductivity) is maintained but the surface property (e.g., wettability) is changed. Given its large surface/bulk ratio, nano-objects (e.g., nanofibers) are particularly amenable to such heterogenous modification. 

However, one of the first reports of CP functionalization, the electrophilic sulfonation of PANI, was carried out with the polymer dissolved in concentrated sulfuric acid, and a bulk reaction occurred. Moreover, the unmodified polymer could be insoluble in the solvent but still be soluble as a functionalized polymer. In this case, the reaction could begin as heterogeneous (surface) but follows as homogenous (bulk). Judicious choice of solvent allows choosing how the reaction occurs and which kind of product is obtained. On the other hand, if bulk functionalization is desired and the product is insoluble, a solvent that wets/swells the polymer is required. Since the diffusion coefficient (Do) inside can be lower (Do < 10^−12^ cm s^−1^) [39] than in solutions (typical Do of ca. 10^−5^ cm^2^ s^−1^ [40], small polymer particle size (the time for mass transport depends on the square of the diffusion path) and high soluble reactant concentration could be used to minimize the effect of slow mass transport inside the polymer. 

The reactions occurring with the main conducting polymer chain are related to the two kinds of monomer units: aromatic rings (including heterocycles) and quinone-like rings. The aromatic rings react via aromatic substitution (with H^+^ as leaving group) and the quinone-like rings by nucleophilic addition. Both types of reactions maintain the extended conjugation related to conductivity. Electrophilic (or radical) addition to ene bonds (e.g., in poly(phenylenevinylene) molecules) will create sp^3^ carbons [41], destroying the extended conjugation. Indeed, this is the cause of air degradation of polyacetylene, for which the addition of water or oxygen causes loss of conductivity. 

The surface of thin films of conducting polymer, often prepared by electropolymerization, can be surface modified changing the surface properties. Additionally, the surface of nano-objects made of CPs (nanospheres, nanofibers, nanoplates, etc.) could be functionalized by the same reactions. In both cases, the methodology resembles the bioconjugate chemistry of biomolecules [42], in which the surface is modified without affecting the bulk. The added functional groups could help stabilize the colloidal dispersion of the nano-objects. 

It should be mentioned that an alternative method of introducing functional groups is the polymerization of monomers containing the desired functional group. First, the synthesis of the functionalized monomer is usually difficult due to the inductive and steric effects of the attached functional group on polymerization. Indeed, it has been reported that anilines bearing strong electron-withdrawing groups (*ortho* and *meta* to the amino group) could not be polymerized [43]. Copolymerization could help produce high-molecular-weight polymers including some amount of functionalized monomer units [44]. However, if the functionalized monomer does not homopolymerize, its reactivity will be lower than the unmodified monomer. Different comonomer reactivity induces other effects, such as low comonomer (of low reactivity) incorporation, formation of block copolymers, and compositional shift [45]. Obviously, homo- and copolymerization of functionalized monomers produce bulk-modified materials and cannot be used to modify the surface of films or nano-objects. The polymerization of previously functionalized monomers [46] requires the functionalization of the monomer while leaving free the positions where the chain formation occurs (at least one N–H and position 4 -*para*- to the amino group in anilines) and both positions to the heteroatom (N in pyrrole and S in thiophene) in pyrroles and thiophene. Therefore, substituents in positions 2, 3, 5, 6 in the ring and 1 in the N could be attached in anilines. In the case of pyrrole, both the N and β positions in the ring can be modified. For thiophene, both β positions (3 and 4) can be functionalized. It should be noted that the α (2 or 5) positions on pyrrole and thiophene and the *para* (4) position in aniline are the most reactive positions due to inductive and steric factors. Therefore, substitution in the monomer through typical reactions (e.g., SEAr) will attach the substituents to those positions, blocking the oxidative polymerization. Therefore, the synthesis of substituted monomers is not simple. Since the mechanism of incorporation of functional groups is quite different from that of postfunctionalization, this kind of work will not be discussed in this review. 

## 2. Covalent Functionalization of Conductive Polymers

The conducting polymers which have been post functionalized are shown in Figure 1. 

In the case of PT, modified polymers made via polymerization of the substituted monomer poly(3-hexylthiophene) (P3HT) have been even more studied than PT. Another substituted CP (poly(3,4-ethylenedioxithiophene), PEDOT) has also been studied and modified extensively. While the modern conducting polymer field began with the discovery of doping in polyacetylene, the low stability of this polymer caused interest to fade out. While all published work is reviewed, the goal of the review is not to simply recount the work but to try to understand the rationale behind the reactions used. Therefore, the results will be divided by type of polymer, with a brief introduction on the organic chemistry nature of the polymer chains. Then, the work in which one type of polymer is modified with the same type of reactions will be discussed. Finally, published research on the functionalization of other conducting polymers for which modifications have been less widely applied will be described for the sake of completeness. In each section, when available, the effect of modification on technologically relevant properties is also briefly discussed. 

### 2.1. Polyaniline Functionalization

Polyaniline (PANI) is the most widely studied CP. It consists of chains of benzenoid rings linked by amino groups (Figure 1). The most stable form is emeraldine in its salt form (PANI(ES), Figure 2), which can be present in protonated form. 

There are two other forms of PANI: the fully reduced form (leucomeraldine, LEB) and the fully oxidized form (pernigraniline, PN). Leucoemeraldine is made of diphenylamine (aromatic amine) units, while pernigraniline consists of quinonimine (non-aromatic but cyclic α,β-unsaturated) units. From the chemical point of view, emeraldine is made of 50% leucoemeraldine and 50% pernigraniline units. Diphenylamine units show typical aromatic reactions (e.g., electrophilic aromatic substitution) while only ortho positions (to any nitrogen) are available. (Figure 3). Any electrophile could attack the aromatic ring, which is activated by the presence of two N: linked to the ring. The N: share its non-bonded electron pair, making the ring more reactive than that in benzene. In some cases (e.g., Cl_2_), Lewis acids could act as catalysts by creating the electrophile. On the other hand, the electrophile could also attack the nitrogen, replacing the hydrogen. Therefore, some reactions (e.g., Friedel–Crafts acylation or alkylation) could give mixed products. The more reactive form of PANI is leucoemeraldine, in which all rings are aromatic. Emeraldine has 50% aromatic rings, showing an intermediate reactivity. Pernigraniline should be unreactive since it has 0% aromatic rings. Some of the reactants (e.g., HNO_3_) are also oxidants and could convert the ES form in PN, inhibiting the reaction. In fact, a sulfonitric mixture of HNO_3_ (65%)/H_2_SO_4_ (98%) in 1:1 relative amounts is usually used to clean up CP films from glassware via oxidation and chain scission. While protective groups (e.g., acetyl) can be used to protect oxidizable amino groups in the nitration of anilines [47], successful nitration of PANI by SEAr has not yet been achieved. Since –NO_2_ groups could be converted into amino, amide, azo, hydrazo, and imine groups, ring-nitrated PANI would be a quite useful precursor for a variety of functionalized PANIs. The reaction of electrophiles in the nitrogen atom is usually reversible, and functionalized polymer chains are recovered only under carefully controlled conditions. 

Moreover, diphenylamine has a N–H group that could be replaced (e.g., used as nucleophiles in aliphatic nucleophilic substitution). Since the non-bonded electron pair in the N is shared with the rings by resonance delocalization, the N–H in diphenylamine units is a poor nucleophile (and base). However, using strong bases (e.g., hydride ion), it is possible to convert the N–H to the amide (>N^−^) ion, which is an excellent nucleophile. Additionally, using activated reactants (e.g., anhydrides), it is possible to substitute the H in the N–H. Both the nitrogen and the ring are susceptible to electrophilic attack. Given the fact that diphenylamine and quinonimine units can be converted between them via oxidation/reduction (both chemical and electrochemical), the reactivity to one type of reaction can be modulated by the oxidation state. Table 1 summarizes the typical reaction in PANI functionalization.

The aforementioned reactions are summarized in Figure 4: 

#### 2.1.1. Electrophilic Substitution (SEAr) on Diphenylamine Rings

##### Electrophilic Aromatic Substitution in the Ring

Sulfonation of Polyanilines

Epstein et al. investigated the sulfonation of PANI (ES form) dissolved in H_2_SO_4_ (c.c., ca. 98%) using SO_3_ provided by fuming sulfuric acid [83]. The product (SPAN) has ca. 50% rings with one –SO_3_H attached. SPAN is soluble in basic media (where the cation radicals deprotonate and the polymer becomes a negatively charged polyelectrolyte). It is insoluble in acid media, where the fixed sulfonate groups compensate (self-dope) the radical cations. In that way, the conductivity does not decrease abruptly at neutral pH, as occurs with PANI. The conductivity is lower than that of the parent PANI (0.1 S cm^−1^ vs. 3 S cm^−1^) due to electronic (charge pinning by the –SO_3_^−^ groups) and steric (hindrance of the bulky sulfonate group with the C–H of the neighboring ring) effects. The self-doping nature allows use of the material in metal (e.g., lithium) rocking chair batteries (in which Li+ produced in the cathode compensates the charge of the sulfonate in the anode), unlike PANI, for which salt is formed during discharge, requiring large solvent volumes [84]. The molecular weight of SPAN was measured using light scattering [30], showing a reduction to that of the parent PANI, implying that the electrophilic sulfonation causes some chain scission. The redox-coupled ion exchange is different than PANI, with two steps of proton expulsion upon oxidation (LE to ES and ES to PN) [12,65,85]. Epstein et al. explored different routes to the sulfonation of ES, including heating with ammonium sulfate and treating with ClSO_3_H, showing that the homogenous reaction with fuming sulfuric acid gives the best results [51]. Chen and Hwang showed the subtleties of self-doping in SPAN [50]. While SPAN is soluble in basic media but not in water, when SPAN (EB) is dialyzed and then converted in its completely self-doped form using an ion exchange membrane, it becomes soluble in water. Sahin et al. prepared sulfonated polyaniline via electrochemical polymerization of aniline in a solution of FSO_3_H/ACN [52]. The FSO_3_H acid act both as electrolyte and sulfonating agent. Up to 89% sulfonation is achieved. Interestingly, the modified polymer is not only soluble in basic aqueous solution but also in NMP and DMSO, revealing that the solubilizing effect of the –SO_3_^−^ group is not exclusive to water. 

Sulfonated polyaniline is likely the most widely used functionalized CP, being applied in anticorrosion coating [86], electrodes for stimulation of biological cells [87], electrochromism [88], interlayer in OLEDs [89], non-covalent functionalization of graphene [90] and carbon nanotubes [91], polymer electrolyte membranes [92,93], cathodes for batteries [94,95], enzymatic electrochemical sensors [96], polymer blends [97], acid catalyst [98], photovoltaic cells [99], composites with hydrogels [100], nanofiltration membranes [101], and stabilizer of colloids [102].

Then, Epstein et al. sulfonated PANI in its leucoemeraldine state, obtaining a polymer with a higher sulfonation degree (ca. 78% of the rings) [53,103]. The polymer (HSPANI) is more soluble, and the conductivity is not affected by the pH (0–14). The conductivity is higher than SPAN, suggesting that is the alternation of sulfonated and unsulfonated rings that pin the charge in SPAN. The XPS study of HSPANI agrees with such a model [104]. Pyskhina et al performed exhaustive sulfonation with HSO_3_Cl, reaching up to 100% sulfonation [70]. Barbero et al. extended the sulfonation methods (from ES and LE forms) to a N-substituted PANI: poly(N-methylaniline) (PNMANI) [56]. Products similar to SPAN and HSPANI were obtained via reaction with fuming sulfuric acid achieving 50% sulfonation (SPNMANI) and ca. 100% sulfonation (HSPNMANI). The redox-coupled ion exchange, measured by PBD, showed proton expulsion in the two redox steps for the HSPNMANI and proton expulsion/anion insertion for SPNMANI [56]. 

Two groups have recently revisited the sulfonation of PANI. Mendes et al. studied different sulfonation routes of PANI [77] concentrated H_2_SO_4_ (100 and 150 °C for 24 h) and fuming sulfuric acid (25 °C for 0.5–2 h). The authors concluded that PANI heated with concentrated H_2_SO_4_ does not become sulfonated because sulfonation is reversible and added groups are released. On the other hand, sulfonation is detected by FTIR when fuming H_2_SO_4_ is used. Using DSC, the authors observe a broad endothermic band in PANI, which they attribute to the hydrogen bonding interactions between neighboring chains. As expected, the band decreases sharply in sulfonated polyaniline since hydrogen bonding is hindered by the steric effect of –SO_3_^−^ substituents. Bhadra et al. sulfonated PANI (ES) for 6 h and obtained a PANI with 93–94% (according to elemental analysis and FTIR) sulfonation (SPANI) [76]. They found via XRD that SPANI is more clearly crystalline (52.4%) than PANI (37.3%) but with wider spacing between chains. A detailed Rietveld analysis of the XRD data suggests that the sulfonic acid groups of two adjacent polymer chains are on the same side and close to each other, likely due to the sulfonate (–SO_3_^−^) acting as a counterion of the radical cations (Ar–NH^+^–Ar). However, only 50% of sulfonates are required because the polymer is in its emeraldine state (50% cation radicals of all N). Therefore, the excess sulfonate could act as a counterion of protonated amine units (Ar–NH_2_^+^–Ar) and/or form intrachain H-bonding (six-member ring structure) between amine groups and sulfonic acid groups of the same polymer chain as well as inter-chain H-bonding between sulfonic acid groups of adjacent polymer chains and amines. The authors found a conductivity of PANI of 0.073 S cm^−1^—low for PANI synthesized in solution (ca. 3 S cm^−1^ [69])—and 0.031/cm for SPAN (lower than that HSPANI ca. 1 S cm^−1^ [69]). However, the ratio SPAN/PANI [R] is ca. 43%, while that of HSPANI/PANI [69] is only 33%. 

The main goal of PANI sulfonation was to produce a conducting polymer soluble in aqueous solution. In fact, SPAN (ca. 50% of the rings sulfonated) shows good solubility in basic aqueous solution but it is insoluble at pH > 5. In neutral–basic solution, PANI chains are deprotonated and the CP is a polyelectrolyte, whereas ion–dipole interactions (of –SO_3_^−^ with water) promote polymer solubility. When the PANI chains become protonated, the sulfonate group compensates the positive charge and the polymer becomes charge neutral. Since the emeraldine base form of PANI is partially (50%) oxidized, 50% sulfonation is required to balance the charge. Moreover, partially sulfonated PANI [51] is only weakly soluble in basic media. The pH is higher than that of PANI(EB) protonation because the negative charges in the sulfonate groups create a Donnan potential, making the internal pH lower than the external pH. On the other hand, HSPANI (75–100% of the rings sulfonated) is soluble even in acid media. There is always an excess (>50%) of sulfonate groups, which allows solubilization of the polymer. The charge compensation process is called self-doping because the polymer contains mobile polarons and fixed counterions. The Donnan potential allows protonation of the quinonimine units and the creation of polarons, which render the polymer conductive. While SPAN is only conductive in acid pH (>7) [48], HSPANI is conductive up to pH 11 [53]. Another property affected is the ion exchange coupled with the oxidation/reduction. The ion exchange in PANI in moderately acidic pH (<2) is as shown in Figure 2. The combination of charge compensation processes (anion insertion on oxidation) with acid–base equilibrium implies that PANI(LEB) inserts two anions into each of the four rings during oxidation to PANI(ES). PANI(ES) releases four protons and four anions, from the tetramer unit, during oxidation to PANI(PNB). Therefore, PANI chains exchange anions during oxidation/reduction.

The incorporation of one sulfonate group per ring changes both redox-coupled ion exchanges (Figure 5):

SPAN (50% sulfonation) will only have a different ion exchange coupled in the first oxidation step. In HSPANI, even the pernigraniline form is completely self-doped. In nonaqueous media (e.g., LiClO_4_/ACN), HSPANI is a cation exchanger, not anion/proton as in PANI. If a battery is built with HSPANI as anode and Li^0^ as a cathode, Li^+^ is produced in the cahode and inserted into the anode during discharge. This is a so-called “rocking chair” battery. On the other hand, using PANI as an anode, anions are produced during discharge in the anode and cations in the cathode. Therefore, large amounts of solvents are needed to store the formed salt in a soluble form [12]. Since the sulfonic group is a strong acid (pKa < 0), SPAN and HSPANI are polymeric acids. Such solid-state acids can be used to build polyelectrolyte membranes [86] or as catalysts for organic chemistry reactions (e.g., transesterification to produce biodiesel) [92].

Bromination (–Br) of Polyaniline

Bromine (Br_2_) is highly reactive towards aromatics (SEAr), resulting in the polybromination of aniline in water [105]. Moreover, p,p′-substituted diphenylamines (such as PANI) give dibromo(ortho)diphenylamine. Stejskal et al. brominated PANI (ES and EB forms) and obtained modified polymers (up to 59% Br/N ratio). The conductivity decreased by ca. five orders of magnitude between PANI and 59% brominated PANI. The ratio can be regulated by the relative amount of Br_2_/PANI. SEAr of Br_2_ on PANI(ES) (or PANI(EB)) should give only 50% bromination since quinonimine rings are not reactive in SEAr. However, since HBr is produced during bromination, it is possible for Br- ions to add nucleophilically to quinonimine units in PANI, explaining the slight excess (59%). While the brominated ring in diphenylamine units would not be reactive enough for SNAr [106], the neighboring quinonimine unit could act as an electron-withdrawing group. Moreover, catalysts (e.g., Cu complexes) could be used to promote SNAr on the PANI–Br moieties [107], making it a possible precursor of a variety of functionalized PANIs. Indeed, pol(2-BrAni) has been used to produce phosphonated polyaniline (Section 2.1.3). Bromination of the formed polymer produces a material with higher electronic conductivity than those produced by homopolymerization of 2-bromoaniline (or its copolymer with aniline). It seems that postfunctionalized polymer chains retain the linear polymer chains of PANI, while polymerization of substituted anilines produced chains with other linking positions in the ring (e.g., *ortho*) that break the extended conjugation. 

Coupling with Diazonium Salts

The electrophilic attack of diazonium ion to aromatics is the most common method of producing azo dyes [108]. The usual site of attack is the *para* position (due to steric constraints in the *ortho* position), but if that position is blocked (as is the case in PANI), the substituent could be linked to the *ortho* position (Figure 6). 

Freund et al. [55] described the reaction of aryldiazonium salts on electrochemically produced PANI. They obtained a nonconductive polymer and proposed a mechanism in which the diazonium ion decomposes, forming the aromatic cation (and nitrogen). The electrophilic cation attacks the nitrogen (see Section “Electrophilic Substitution in the N–H Group”), forming a N-aryl-substituted PANI with triphenylamine units. The modified polymer is electro-inactive, analogously to the polymer produced by oxidative polymerization of 4-(phenylamino)benzenesulfonic acid [109]. The conditions used (acid media and room temperature) favor degradation of the diazonium ion. On the other hand, using the typical reaction conditions for the diazonium coupling—basic/neutral media and low temperature (0–5 °C)—together with a N-substituted-polyaniline (poly(N-methylaniline) allows the coupling of the diazonium ion with the polyaniline chain, forming azo bonds [110]. The presence of azo (–N=N–) groups is confirmed with FTIR. Since p-aminobenzensulfonic acid was used to make the diazonium ion, the polymer is soluble in basic media. On the other hand, thin films of the modified polymer, deposited from solution, are insoluble and electroactive in acid media, unlike the polymer made by DeArmitt et al. [109], suggesting that the substitution occurs in the ring and not in the nitrogen. This is expected since substitution in the nitrogen of PNMANI involves removal of the methyl group. However, the procedure was extended under the same coupling conditions (buffered solution of pH = 8, temperatures below 10 °C) by Acevedo et al., to the functionalization of PANI [111]. Both reaction mechanisms are shown in Figure 6. While the cyclic voltammograms of the modified polymer differs to that of PANI, no loss of electroactivity is observed. Moreover, the effect of functionalization on the CV is similar to that observed in polyanilines bearing electron-withdrawing groups linked to the ring [112]—that is, a decrease in the potential difference between the PANI(LEB)-PANI(ES) and PANI(ES)-PANI(PNB) redox transitions. The data differ from those described by Liu and Freund [55], who found electrochemical deactivation upon treatment of PANI films with aryldiazonium salts. However, the reaction conditions (acid media, ambient temperature) promote the decomposition of the diazonium ion to form the aryl cation, which attacks the nitrogen. 

Different diazonium salts produce similar modifications. The modified polymers suffer degradative reduction with dithionite where the azo (–N=N–) group is converted into amino (–NH_2_) groups. Therefore, a PANI bearing amino groups in the ring is produced. The redox response is more similar to the one of unmodified PANI than when azo groups are attached, likely due to the stronger inductive and steric effect of the azo group compared to the amino group. The amino group could be converted into a variety of functional groups (e.g., amide), but such developments have not been pursued. It is noteworthy that emeraldine base was used as reactant, and diphenylamine rings are the only reactive moieties towards diazonium ion attack. On the other hand, using LE as reactant will allow the modification of every ring in PANI. Since the reaction is quite straightforward (aqueous solution, low temperature) is amenable to combinatorial functionalization. Different aromatic amines can be diazotized and coupled with PANI. Additionally, azo dyes can be made via reaction of diazonium salts with other primary aromatic amines. In that way, azo dyes which can be diazotized into diazonium ions to react with PANI can be produced. By changing the reactants (primary aromatic amines, Figure 5) of the azo dyes, single linkage combinatorial modification can be achieved. Those coupling agents (CA) are primary aromatic amines (e.g., 4-aminobenzensulfonic acid) which can form diazonium ions able to couple with other activated aromatic amines but cannot couple with themselves or be attacked by diazonium ions. On the other hand, those called aromatic amines (AA) can couple with other AA (or themselves) and be attacked by CAs. Therefore, there are AAn-AAn dyes but not CAn-CAn. The reason is that CAs bearing electron-withdrawing groups (e.g., –NO_2_) are unreactive with a weak electrophile as the diazonum ion. On the other hand, the presence of electron withdrawing groups in the ring destabilizes the positive charge of the ion, making it a better electrophile. To the diazotized dyes are added the diazonium salts of CAs and AAs. 

Using just 6 aromatic amines (Figure 7), it is possible to create 17 functionalized polymers. Moreover, using aromatic primary diamines (e.g., benzidine), it is possible to link any activated aromatic ring (e.g., 2-naphtol) with a primary aromatic amine (the diamine itself or another primary amine (e.g., 2-aminobenzoic acid) which can be made into a diazonium ion and coupled with PANI. In the work [113], 65 functionalized polymers were produced, which constituted a large part of the known functionalized PANIs (Figure 1). The modification efficiency depends on nature of the diazonium ion and it being in the range 7–33%.

The target properties were two: conductivity and solubility in common solvents. A high-throughput screening procedure was developed where thin films of PANI were chemically polymerized onto thin PE films. The thin (ca. 2 μm) weakly absorbing substrate allows the measurement of the UV–visible and FTIR spectra of the conducting polymer films (ca. 200 nm) via transmission and the measurement of conductivity [114]. In that way, the success of the modification can be checked. By painting two contacts (silver ink) at a known distance, the conductivity can be measured. As expected, all modified polymers have lower conductivity than PANI due to electronic and steric effects. The modification with diazonium ions from azo dyes produces lower conductivities than with those diazonium ions from simple aromatic amines. By immersing a known area of supported film in different solvents, the solubility can be assessed by the removal of the film from the substrate. With those showing solubility, it was measured as highly soluble (HS > 1% *w/v*) or soluble (1%> S > 0.1% *w/v*) [115].

Polymers functionalized with charged groups (–COO^−^, SO_3_^−^) show good solubility in basic aqueous and alcoholic media. Unexpected results were that the polymers modified with nitro (–NO_2_) or even azobenzene groups (PANI-AA1-AA1) showed high solubility in toluene, chloroform, and acetone. 

The main goal of coupling with diazonium salts was to improve the solubility of PANI in different solvents while decreasing polymer conductivity as little as possible. Using a combinatorial approach to both synthesis and property screening allows the identification of conducting polymers with high solubility in low polarity organic solvents (e.g., toluene) via modification with azo dyes bearing nitro groups. The effect is difficult to predict from structure–property relationships. The solubility allows the coating of solids using those materials and the blending of the conductive polymer with dielectric polymers (e.g., polystyrene). Using azo dyes as added functional groups incorporates other properties, such as additional absorption bands in the visible spectrum [115], photochromism [115], and even the possibility of selective insolubilization of the functionalized polymer by localized reductive degradation of the azo group [111]. PANI chains modified with azo moieties bearing ionic groups (e.g. –N(CH_3_)_4_^+^) are soluble in water and have aromatic rings (both in PANI and CA). Therefore, they can form π–π interactions with the aromatic ring in nanocarbons. Allowing its use to stabilize dispersions of multiwall carbon nanotubes in water [116], and likely to occur with other nanocarbons (graphene, single wall carbon nanotubes). The dispersed particles bear net charges and can be assembled electrostatically in layer-by-layer multilayers [116]. Moreover, the ionic group could be locally removed via reductive degradation [111] to produce 2D patterns of the nanocarbon or removed from the whole multilayer to make it resistant to dissolution. 

##### Electrophilic Substitution in the N–H Group

Amide Formation at the Nitrogen

The first functionalization of PANI was achieved by Wrighton et al. [58]. They produced PANI films electrochemically and made them react with different anhydrides (in acidic ACN). The reactivity depends on the oxidation state of PANI. The reactivity goes in the order: (F_3_CCO)_2_O > (C1_3_CCO)_2_O > (H_2_ClCCO)_2_O > (HCl_2_CCO)_2_O > (H_3_CCO)_2_O. Lin and Chen reacted PANI (EB dissolved in NMP) with sulfobenzoic anhydride [59]. The anhydride formed an aromatic amide with the diphenylamine nitrogen (Ar–NH–Ar), leaving the sulfonate group linked to the benzene ring and able to self-dope the PANI chain. The functionalized polymer is soluble in water and has a conductivity of ca. 7 × 10^−4^ S cm^−1^. Barbero et al. reduced PANI(EB) to PANI(LE) with phenylhydrazine to increase the number of reactive N–H groups [67]. Then, they reacted PANI(LE) with neat ethanoic anhydride at 80 °C. A high (>80%) modification ratio can be achieved, and the product is soluble in common solvents (e.g., CH_2_Cl_2_). However, the polymer has very low conductivity and no electrochemical activity. It seems that substitution in the nitrogen with electron withdrawing groups (e.g., –COR) of PANI decreases the conductive and redox properties. Liu and Freund reacted aryldiazonium salts with PANI films (electrochemically produced) in acid media and ambient temperature [55]. The diazonium ion decomposes to form a highly reactive (non-aromatic) aryl cation which substitutes at the nitrogen, forming triphenylamines. The PANI film loss electroactivity on a degree related with the reaction time, suggesting that the reaction occurs from the film/solution interface inwards. The resulting polymer is similar to that produced by DeArmitt et al. via the polymerization of diphenylamines [109]. Both polymers are electroinactive due to the steric effect of the bulky aryl substituent, which hindered the planarity of the rings in the PANI chain, decreasing extended conjugation. 

Wrighton et al., modified electrochemically produced PANI films supported on a special three electrode device, which acted as a transistor [58]. Therefore, the goal of the modification was to modify the electronic properties of PANI. It was found that in situ modification in the amine nitrogen drastically decreases conductivity. A similar effect was observed by other researchers [55,67]. While the production of a whole non-conductive film is of little interest, controlled functionalization of the external surface (e.g., making it hydrophobic) could produce transistors with sensitivity to different chemical entities (e.g., fatty acids) than PANI. Moreover, localized formation of non-conductive domains (e.g., by mask-controlled reaction) could allow the drawing of conductive patterns in non-conductive films to produce flexible electronic devices. 

Tertiary Amine Formation at the Nitrogen

Hua and Chen produced a self-doped PANI substituted in the diphenylamine nitrogen [54]. Since the diphenylamine is a poor nucleophile, they first reacted PANI (EB) with sodium hydride to form the amide (Ar_2_N^−^). Then, they reacted the amide with p-(3-BrC_3_H_6_)–C_6_H_4_–SO_3_–Na^+^. The amide attacked the alkyl halide (SN_2_, forming a new amine with a pendant group containing sulfonate. Therefore, the cation radical of PANI backbone was self-doped by the sulfonate groups. The polymer is soluble in water and has a conductivity of ca. 2 × 10^−2^ S cm^−1^. Barbero et al. used SNAr reaction to modify PANI(LE) [67]. The reaction was performed with activated arylhalides (bearing –NO_2_ electron-withdrawing groups). The yields were low since diphenylamine is a poor nucleophile. Raffa et al. reacted PANI (LE) films (prepared electrochemically) with propanesultone [78]. The ring tension in propanesultone made it prone to nucleophilic attack by the N–H groups in PANI (LE). The sultone ring was opened, and PANI became N-alkylated with a pendant sulfonate group. Therefore, the PANI chains became self-doped and were electroactive in neutral pH (unlike PANI), allowing the construction of an enzymatic sensor active at pH = 7. Incorporation of ionic groups (–SO_3_^−^) attached to PANI chains makes the polymer electrically conductive at neutral pH with a modest loss of conductivity. It should be remembered that PANI shows a five-orders-of-magnitude loss of conductivity when exposed to neutral media. This property of functionalized polymers allows different biological applications, such as in electrodes for sensors/actuators. Indeed, Raffa et al. [78] produced an enzymatic electrochemical sensor using such an approach. 

Reversible Formation of Nitrosamine

Salavagione et al. showed that nitrosonium ion (NO^+^), which is formed via the protonation and dehydration of nitrous acid, electrophilically attacks the nitrogen of the diphenylamine units, forming nitrosamines [80]. The nitrosamine of PANI is soluble in common solvents (e.g., chloroform) and can be cast into films. However, it is unstable in acid media, losing the NO and reverting to PANI. The reversible process was used to create conductive patterns with chemical lithography using a spray of acid. In a more detailed study, the authors studied the formation and decomposition of the nitrosated PANI. Then, using a photoacid generator (PVC), it was possible to create conductive patterns by photolithography of nitrosated PANI. The reversible nature of this postfunctionalization, coupled with the solubility of nitrosated PANI in nonaqueous solvents, allowed the drawing of 2D patterns of PANI on surfaces, which can be used in flexible electronics. 

#### 2.1.2. Nucleophile Addition on Quinonimine Units 

While other reactions of CPs (e.g., SEAr) are not exclusive of CPs and could be used to modify other polymers (e.g., polystyrene [117]), nucleophilic addition involves α,β−unsaturated bonds, which are the backbone of the extended conjugation, giving CPs their electronic properties (Figure 8) [118]. Therefore, it is a reaction intrinsic to CPs. 

Han and Jeng discovered that a solvent of PANI (pyrrolidine) react with PANI by nucleophilic attack [61]. They then observed a reaction with thiols (dodecanothiol and mercaptoacetic acid). They observed that reaction with the nucleophile produces the reduced form of PANI (a functionalized LE) and named the phenomena “concurrent reduction and substitution reaction”. They extended the method to mercapropanesulfonic acid [62], producing a self-doped polyaniline with higher conductivity than SPAN. Interestingly, they used acetic acid as a catalyst. 

Salavagione et al. synthetized a self-doped polyaniline (functionalized with –SO_3_^–^ groups) via nucleophilic addition of sulfite ions to emeraldine salt and pernigraniline [64]. It was already found that the reactivity depends on the relative amount of quinonimine units in the polymer. Therefore, PN is more reactive than ES, and LE is unreactive. At the same time, it was found that decreasing the pH increases the reaction rate since protonated quinonimine is more reactive to nucleophilic attack than deprotonated attack. Accordingly, bisulfite is more reactive than sulfite. Obviously, this depends on the nucleophile since amines would be protonated by the acidic pH. Moreover, the so-called “concurrent reduction” of Han et al. [61] is a result of the addition mechanism, which gives the LE form of the functionalized PANI as product. If no oxidant is present, the reaction will stop when all (50% of the aniline rings) are converted. However, an additional oxidant can be used to reconvert the amine to quinonimine units. In addition to sulfite ions, related arylsulfinic acids were used as nucleophiles. Finally, a complete set of nucleophiles, which included acetoacetate and cyanide, were used to modify PANI. Kumar et al. prepared PANI polymer brushes linked to the electrode by thiolate bonds [72]. Then, they reacted PANI short chains with sulfite, changing the voltametric response. In that way, an integrative sensor of sulfite was developed. However, given the reactivity of quinonimine with biologically relevant molecules, such as thiols and amines, the sensor could not be used in biological matrixes or food (e.g., wine). It is noteworthy that the nucleophilic addition of sulfite ions produces sulfonated polyaniline with different degrees of sulfonation (from 25% to 100%) in an ecofriendly way (using water or ethanol as solvent) without corrosive/toxic reagents (e.g., fuming sulfuric acid). Moreover, the less harsh conditions (50–70 °C at pH = 3) avoid chain scission, as was shown by measuring the molecular weight of the polymer during reaction [87]. However, it has seldom been used to produce sulfonated polyaniline in the myriad of applications. 

Han and Chen modified PANI through nucleophilic addition of fluoride ions in MeOH [119]. They used reoxidation as a way to increase the substitution degree of PANI from 25% (1 F per 4 rings) to 125% (5 F per 4 rings). It was observed that EB only produces 25% substitution while PNB produces 50%. Then, they reoxidated the functionalized monomer in suspension electrochemically (ACN/LiClO_4_). However, the authors took great pains to deprotonate the functionalized PANI, although it has been shown that protonation increases the reactivity. Han et al. [120] used the technique to functionalize PANI and obtain a butylthio-functionalized derivative which has higher conductivity than the unmodified PANI. Han et al. used the butylthio-modified PANI to produce nanospheres, which were then converted to graphitic nanospheres [121]. Levon et al. used nucleophilic addition to modify PANI (a film produced electrochemically) via nucleophilic addition of 2-mercaptoethanol [69]. Then, they attached a redox group (ferrocene) by linking it with the –OH group. Laiff et al. used nucleophilic addition of thiols to modify the surface of PANI nanofibers [122,123,124]. The nanofibers, functionalized with carboxylic or amide groups, were used to immobilize biomolecules [125], including redox enzymes which were electrically connected to the conducting nanofibers. Interestingly, the nanofibers whose surface is modified with self-doping groups show electroactivity at neutral pH and can be used to electrochemically detect ascorbic acid [126]. This result confirms the model proposed before [127], which is interparticle resistance at high pH, which inhibits PANI electroactivity but not bulk effects. Bongiovanni Abel et al. [26] modified PANI chains via the nucleophilic addition of cysteamine. The thiol group is a better nucleophile and became attached to the PANI chains. Then, dansyl chloride was reacted with the aliphatic amino groups, rendering the PANI fluorescent. The extended cysteamine linker avoids quenching of the naphthalene rings by the quinonimine units of PANI. Fluorescent PANI NPs are made from the solution of the modified PANI chains in NMP via solvent displacement with PVP as stabilizer. 

Yslas et al. modified the surface of PANI via NCA of the aminiacid cysteine and made the film compatible with two biological cell lines [128]. Neira-Carrillo et al. modified PANI nanoparticles via NCA with cysteine [129]. The –COOH groups in cysteine induce the crystallization of calcium-carbonate, producing biomaterials with photothermal activity. Durgaryan et al. observed the addition of thiosulfate ions [74] and hydrazine [75] to a polymer (produced by oxidative polymerization of p-phenylenedianine) with a structure very similar to that of pernigraniline. It is likely that PN will give similar reactions with those molecules and the related hydroxylamine and phenylhydrazine. Recently, Amaya et al. investigated the phosphonilation of PANI in solution (NMP) by nucleophilic addition of p(OEt)_3_ [130]. To improve the functionalization degree, they reoxidized the partially modified PANI with persulfate ion, in line with the mechanism of nucleophilic addition. However, they still used the terminology of substitution. 

Nucleophilic conjugate addition (NCA) allows the incorporation of a variety of functional groups in the bulk or on the surface of films (or nanoparticles) made of PANI. Again, attaching ionic groups (e.g. –COO^−^) increases bulk polymer solubility while creating pH dependent solubility. In acid media, the functionalized polymer is conductive and electroactive but solubilizes in basic pH. As a solution, it can be used as ink in common printing techniques (e.g., inkjet printing) to produce flexible electronics or electrochemical sensors. The same effect is used to produce stable dispersions in water of PANI nanoparticles whose surfaces were modified via NCA. In both cases, using functional groups which can be protonated/deprotonated allows control of the solubility. In that way, films of modified PANI with carboxylic (–COOH) groups can be deposited from ammonia/water solution (bearing –COO^−^NH_4_^+^ groups), which—upon drying and gentle heating—revert to the acid form and become insoluble. Moreover, using a photoacid generator, patterns of insoluble conductive polymer can be drawn photochemically. In the case of PANI films, only the surface can be modified. The functionalization of the surface of films or nanoparticles via NCA allows changing of the interaction of the solid with biological entities. In that way, the biocompatibility was improved [128] or an inorganic biomaterial was synthesized [129]. In the latter case, the molecule attached through NCA (cysteine) induces the formation of the inorganic biomaterials (crystals of calcium carbonate), which are biocompatible due to the inorganic matrix, while the PANI nanoparticles absorb NIR light, allowing its use in photothermal therapy. Other properties incorporated into the surface of PANI entities through NCA are additional redox groups (ferrocene) to films [69], fluorescence to nanoparticles [26], and biomolecules to nanofibers [121]. 

#### 2.1.3. Reactions on Preattached Reactive Groups 

Since PANI has only some reactions in the polymer backbone, it is possible to produce PANIs with attached reactive groups (by oxidative polymerization or copolymerization). The approach has not been used as much as in the case of polythiophene, but some reactions have been studied. Amaya et al. used Pd^0^ as a catalyst to promote the S_N_Ar reaction of HPO_2_H_2_ with poly(2-bromoaniline) (produced by oxidative polymerization of 2-bromoaniline) [71]. In that way, self-doped phosphonate polyaniline was produced. Shoji and Freund produced a substituted PANI through the electrochemical polymerization of 2-aniline boronic acid in presence of fluoride [73]. Then, the boronic acid moiety was converted into –OH or –X (halogen). The poly(2-hydroxyaniline) could not be produced by polymerization of 2-hydroxyaniline (2-aminofenol), which produced a ladder polymer [131]. 

Salavagione has copolymerized (chemically and electrochemically) aniline with 3-ethinylaniline to produce PANI bearing alkyne groups [79]. The modified PANI could react with thiols through the “click” reaction thiol-yne [132]. In that way, very fast functionalization of PANI with a variety of functional groups could be produced. A preliminary test with PEG-dithiol shows that the copolymer is able to link to the PEG chains in a fast way. Using NCA, it is possible to attach nucleophiles (e.g., thiols) which bear active groups. In that way, an electroactive group (ferrocene) was attached to the surface of electrochemically produced PANI films [69]. Moreover, the surfaces of nanospheres were modified with cysteamine and a fluorescent group attached to them. The alkyl chain act as spacer separating the fluorophore from the quinonimine units and limiting the redox quenching [26]. Finally, a thiol (nucleophile) bearing a long (C_10_) alkyl chain with an active group (–COOH) at the other end is attached to PANI nanofibers [121,122,123]. To the carboxylic group, biomolecules (e.g., redox enzymes) are linked by protein bioconjugate reactions. In that way, electrochemical enzymatic sensors can be built [125]. 

### 2.2. Polythiophene Functionalization

Polythiophene (PT) and its derivates are another class of conducting (π-conjugated) polymers that allow chemical modifications similar to what was afore-described for PANI and PPy. The basic structure of PT is represented in Figure 9. As can be seen, this thermally stable CP has the possibility of doping, which leads to electron delocalization of π-orbitals along the polymer backbone, giving optical and electronic interesting properties [133]. The remotion of electrons of PT conduces to the formation of polaron/bipolaron depending on the number of removed electrons (one or two electrons) [134,135].

However, in the last three decades, a wide variety of research evidenced the versatility of PT for chemical functionalization post-polymerization by using several kinds of organic reactions. Among others, halogenation, cyanation, azide functionalization, esterification, alkyl, and aryl substitution on the PT backbone were described in the literature. Moreover, interesting modifications employing ferrocene, reduced-graphene oxide (RGO), Si surfaces, and combination with other organic molecules were developed for several kinds of applications. In the following paragraphs, the most representative and interesting functionalization procedures for PT are enlisted (Table 2) and described.

#### 2.2.1. Direct reactions on the Thiophene Ring

##### Nucleophilic Addition

In PT the so-called “overoxidation” process is a thoroughly studied phenomenon [155]. In fact, PT is not stable at the redox potentials used for its synthesis [156]. Early on, it was recognized that the degradation reaction involves the nucleophilic attack of water (or OH–) on the oxidized form of the polymer, giving a ring substituted by –OH, which oxidizes to >C=O [157]. Based on that information, Pickup and coworkers studied the addition of nucleophiles (Cl–, Br–, CH_3_O–) to electrochemically oxidized PT, P3MeT, and poly(2,2′-bithiophene) [136]. While they show experimentally clear nucleophilic addition on the quinonoid form, producing the modified and reduced form, they discuss the phenomena as substitution. The mechanism proposed to explain the experimental data (Figure 1 in Reference [136]) seems to be incorrect since it shows the oxidation (−2 e^−^) of the product of the addition to the aromatic form (which is the reduced state). It is found that reaction does not occur with I^−^. This is surprising since I^−^ is a stronger nucleophile than Cl^−^ or Br^−^. However. at +1.4 V_SSCE,_ I^−^ converts into I_2_, which is not nucleophilic. 

The electrochemically driven halogenation of PT derivatives was extensively studied by Inagi et al. [137]. Initially, the authors assumed that the halide is converted into the halogen, which electrophilically attacks electrophilically the polymer. However, the film is in the same electrode (anode) in which the halide oxidation should occur. Therefore, it is oxidized to quinonimines which are not reactive in electrophilic aromatic substitution (SEAr). The authors carried out a detailed study of the reaction mechanism, showing that elemental halogens are unable to react in the conditions of the experiment, while conjugate nucleophilic addition (NCA) is most probable mechanism [158]. A poly(thiophene-alt-fluorene) synthesized via Suzuki–Miyaura coupling polymerization was exposed to tetraethylammonium chloride (Et_4_NCl) in acetonitrile, and electrochemical halogenation was made during constant-current electrolysis [138]. The chlorination efficiently and selectively occurred at the thiophene ring as revealed by EDAX and ^1^H NMR. When halogenation was attempted to introduce bromine or iodine, the functionalization degree found was less than chlorine (for bromine) and did not occur for iodine. This is the same order previously found by Pickup et al. [136] for nucleophilic addition. The order of nucleophilicity is I^−^ > Br^−^ > Cl^−^ > F^−^ [159], but the order of electrode potential (for the conversion of halide into halogen) is F^−^ > Cl^−^ > Br^−^ > I^−^ [160]. At the high potential used (1.65 V_NHE_), I^−^ is converted into I_2_ and no reaction occurs. The lower reactivity of Br^−^ vs. Cl^−^ could be due to steric effects since the attack is ortho to the N–H group. 

The same halides also attach to poly(3-hexylthiophene) P3HT. The halide attach the halide atom to the 4-position (only free) of the repeating 3HT ring [139]. In more recent work, Inagi et al. compared the chlorination of P3HT to the selenophene-containing (co)polymers –P3HS and P3HT-*b*-P3HS- [140]. The results indicated different degrees of chlorination for each polymer. For homopolymers, P3HS resulted in 65% of chlorination, whereas P3HT reached ca. 49%. In the case of P3HT-*b*-P3HS, 49% of selenophene rings were chlorinated compared to the 30% of thiophene rings for the P3HT segment, which was attributed to the crystallinity of the polymer. Moreover, when the process was performed on a statistical copolymer (P3HT-*s*-P3HS), a similar trend was observed regarding the selectivity of sites for Cl^−^ attack. The use of a boron trifluoride-diethyl ether (BFEE) instead of acetonitrile as an electrolyte during the chlorination highly improved the chlorination of P3HT. This fact can be explained as being due to the decreases of oxidation potential of P3HT in BFEE [161]. Additionally, it was demonstrated that Lewis acids (e.g., AlCl_3_) can facilitate the nucleophilic addition of Cl^−^ to P3HT (achieving higher chlorination degree) and other π-conjugated polymers [141]. As in the case of BF_3_, the Lewis acid dopes the CP and decreases the potential for formation of the quinone-like species, which react with the nucleophiles. Additionally, AlCl_3_ increases the stability of P3HT because it reacts react with water, which will act as a nucleophile, degrading the polymer. Other PT derivatives such as poly(thiophene-alt-9,9-dioctylfluorene) (PTF) and poly(bithiophene-alt-9,9-dioctylfluorene) (PBT) were successfully functionalized via nucleophilic addition of halide using a thin-layer cell [142].

##### Electrophilic Aromatic Substitution (SEAr) on Polythiophenes 

Li et al. functionalized P3HT using SEAr. They produce polymer chains with –Br (using NBS), –Cl (using NCS) and –NO_2_ (using fuming nitric acid) as substituents in the 4-position of the thiophene rings [143]. Then, the –NO_2_ substituents, (made by the same method) were converted to –NH_2_ by reduction with iron powder in water. In this way, the use of any catalyst or additives is avoided [162]. This methodology is considered eco-friendly, nonhazardous, and cheap, among other advantages. However, it is surprising that a –NO_2_ group linked to a solid reacts with another solid (Fe). It seems likely that local corrosion of solid Fe generates Fe^2+^, which is the real reductant and is oxidized to Fe^3+^ by the –NO_2_ groups. The solid Fe could then reduce Fe^3+^ to Fe^2+^ in a shuttle mechanism. The –NH_2_ group is then converted to diazonium ion (–N_2_^+^) through reaction with nitrite. Then, using Sandmeyer’s reaction, the diazonium ion is substituted with –CN using copper (I) cyanide [144]. PT functionalization through electrophilic bromination was also explored. For example, a post-polymerization strategy to functionalize the 4-position of P3HT was reported [143,145]. The Br–P3HT was obtained with a high degree of bromination through electrophilic reaction of the polymer solution in chloroform via the addition of N-bromosuccinimide (NBS). In other studies, the authors suggested that bromination of P3HT disturbs the delocalization of p-conjugated electrons on the P3HT backbone. Different bromination degrees (2%, 11%, 22%, 37%, 66%, 84%, and 100%) of P3HT were correlated with the delocalization of π-electrons in the polymer (intra- and interchain), which affect the target properties of the conducting polymer (e.g. photovoltaic properties) [146]. In the same way, recently, some authors employed brominated PT derivates, highlighting the importance of the feasibility of the reaction as well as the great potential of the products to build hybrid materials with potential applications in energy and solar cell devices [163].

##### Substitution of Lithiated Thiophene Rings

The use of lithiated heterocyclic rings (made via reaction of halogenated aromatics with lithium) is a useful strategy to produce a variety of substituted heterocycles [164]. Swager and coworkers functionalized P3HT via lithium–bromine exchange for the introduction of fluorine on the 4-position [145]. The fluorination was significantly higher (ca. 67% of fluorination degree) compared to other reported monomer-modification approaches. The functionalization was easily made as follows: (i) lithiation of P3HT via reaction with n-BuLi/THF and (ii) reaction with N-fluorobenzenesulfonimide (NFSI), corroborating the modification using ^19^F NMR. Another interesting example of fluorinated PT derivate obtention is the inclusion of pentafluorobenzene (PFB) as an end-group in poly(3-octylthiophene) (P3OT) via in situ quenching of the polymerization [147]. The resulting functionalized polymer allows reaction via nucleophilic aromatic substitution (SNAr) with common nucleophiles (e.g., thiols, alcohols, amines, etc.) in mild conditions as well as the synthesis of diblock polymers. 

#### 2.2.2. Reaction with Active Groups Present in Substituted Polythiophenes

##### Azide Moiety and Reaction with Alkynes 

PT containing azide-functionalized chains can be typically prepared using strategies that involve nucleophilic substitution of haloalkyl side chains with azide anions and also by hydroxyalkyl chain azidation (e.g., Mitsunobu reaction) [148,165]. However, in order to minimize the step reaction number and improve the yield, new strategies were developed. For example, Nam *et al.* synthesize poly(3-hexylthiophene-stat-3-(6-azidohexyl)thiophene) (P3HT-N5) by the polymerization reaction of a tert-butyldimethylsilyl (TBDMS)-protected thiophene with subsequent desilylation and conversion of the alcohols to azides [148]. First, regioregular random poly(3-hexylthiophene-stat-3-(6-(tert-butylmethylsilyloxy)hexyl)thiophene) (P3HT-Si5) was synthesized by Grignard metathesis polymerization and then P3HT-OH5 was obtained by remotion of TBDMS using tetrabutylammonium fluoride. Finally, the –OH groups were transformed to azide groups using Bu_4_NN_3_ in presence of triphenylphosphine obtaining the product (P3HT-N5). For the functionalization of azide terminal moieties on PT derivates, several reports have been available in the literature. 

##### Amide Functionalization of Carboxy Substituted PT

The incorporation of pendant dianiline groups on PT to improve the solution processability can be achieved by a series of reactions. First, the copolymerization of alkylthiophene with acetate-functionalized thiophene is obtained, followed by the hydrolysis of the carboxylic groups. Then, via amidation reaction with *N*-phenyl-1,4-phenylenediamine, the oligoaniline groups are attached to the PT backbone, as was confirmed by NMR, FTIR, and elemental analysis [150]. 

Zotti et al. reported PT-based copolymers obtained by anodic oxidation of 3,4-diamino- or 3,4-dinitro-terthiophenes [166]. In this form, the –NH_2_ or –NO_2_ moieties are directly linked to the conjugated backbone. The postfunctionalization of the polymeric films allows or the conversion of the –NO_2_ to -amino groups by reduction with SnCl_2_ in ethanol/HCl. Additionally, 3,4-diamino-2,2:5,2-polythiophene and 3,4:3,4-bis(ethylenedioxy)-3,4-dinitro-2,2:5,2 -polythiophene are able to react with glyoxal in ethanolic solution by condensation obtaining pyrazine rings. The postfunctionalization of the PT-based films provokes important changes from the point of view of the electronic, including optical and electrochemical properties, allowing modulation of them. Other strategies for the synthesis of a variety of aminoalkylsulfanyl PT presenting water solubility were reported as a result of copolymerization via Stille coupling [167].

##### Anionic and Cationic Moieties

In the last decade, the generation of cationic and anionic PT derivatives has been explored. P3HT can be functionalized by attaching a carboxylic acid moiety (P3HT-COOH) through formation of the carbanion, followed by the reaction with carbon dioxide [151]. On the other hand, cationic functionalization of P3T can be achieved via a series of reactions [46]. First, allyl-functionalized P3T was obtained through reaction of the carbanion form of P3T with 3-bromopropene. Second, P3T-Br was synthesized via catalyzed hydrosilation reaction of the terminal double bonds present in P3T-allyl with 4-bromobutyldimethylsilane. Finally, the polycationic-methylimidazolium-functionalized, branched P3T-MIM was synthesized by a nucleophilic substitution reaction of P3T-Br with N-methylimidazole. This imidazolium-functionalized product is promising for several technological applications due to its role in forming ionic liquids. Furthermore, the authors highlighted the versatility of the synthetic route to tune the alkyl spacer length and also the feasibility of using different nucleophiles in order to obtain other classes of end groups. In a similar manner, So et al. comprehensively reviewed the novel methodologies for the synthesis of a wide range of cationic PT through approaches that involved branched polymers, polyelectrolyte diblock copolymers, and post-polymerization functionalization of regioregular poly(alkyl)thiophenes derivatives, among others [46]. An interesting work reported the synthesis of cationic water-soluble PT grafted to reduced graphene oxide (RGO) sheets. The procedure involves the quaternization of RGO-*g*-P3BPT using trimethylamine for the formation of poly(3-(3′-thienyloxy)propyltrimethylammonium bromide) [168]. The synergistic combination of the functionalized conducting polymer and the RGO is promising in the field of antibacterial photothermal therapy. Another method involved the crosslinking of modified PT-ester with branched polyethylenimine (PEI) via ester aminolysis reaction under mild conditions in N-methyl-2-pyrrolidone (NMP) [152]. These cationic, water-soluble PEI-crosslinked-PT derivates were used to build Cu^2+^ sensors. The synthetic methodology is an alternative method to the conventional conversion of conjugated polymer with ester group to water-soluble polymers through hydrolysis under basic conditions [169].

##### Reaction with other Carboxylic Functionalities

Using poly[3-(6-bromohexyl)thiophene] (P3T-6BrHex) as a start polymer, the synthesis of a PT containing an ester group can be achieved. The method implies a single step for functionalization using sodium hexanoate in DMF as solvent, and the final functionalized polymer presents remarkable microstructural features [153]. Other researchers compared the obtention of a PT derivative containing ester moieties (poly[3-(4-butanoyloxy) butylthiophene]) through two different synthetic routes: the direct oxidative polymerization versus the postfunctionalization of poly[3-(4-bromo)butylthiophene. While the nucleophilic substitution of sodium butyrate in the Br atom of the polymer resulted in excellent reaction yield, the polymerization of 3-(4-butanoyloxy)butylthiophene monomer presents the advantage of versatility for the generation of other functionalized polyalkylthiophenes [170]. Through the covalent grafting of arenediazonium tosylate containing –COOH groups with (poly(3,4-thylenedioxythiophene)-poly(styrenesulfonate) (PEDOT:PSS), the introduction of the carboxylic moiety on the CP can be made [171]. In this case, the diazo-cation acts as a counterion of the sulfonic groups (cationic exchange), allowing the modification of the pristine PEDOT:PSS thin films, changing its electrical and surface properties. More recently, the covalent coupling of poly(3-thiophene acetic acid) (P3TAA) with a C-protected tripeptide was also reported to produce a bioconjugated conducting polymer [172]. Through alkaline hydrolysis, a carboxylic-acid-deprotected tripeptide polythiophene conjugate can be obtained [173]. Istif et al. produced aldehyde derivatives of thiophene [172], which are polymerized into conductive polymer chains with aldehyde pendant groups. The high reactivity of aldehydes allows producing conducting polymers with a variety of functional groups attached to the chains. In addition to the functional groups linked to PT chains, semiconducting oligomers of aniline can also be attached [174,175]. A thiophene monomer, substituted with alkyl chains terminated in carboxyl (–COOH) groups is amidated with amine (–NH_2_) terminated oligoanilines. While the conductivity of the polymers is not as large as could be expected of a 3D network of conductive moieties, the physicochemical properties of both conducting chains are observed. Bauerle described work on post-polymerization functionalization of conducting polymers (poly(alkylthiophene)s) substituted with replaceable ester groups [176]. The electrochemical properties of the films can be tuned through the incorporation of different attached groups (e.g., ferrocene). 

##### Reactions with Attached Hydroxyl (–OH) Groups

Poly[3-(10-hydroxydecyl)thiophene] can be synthesized via alkaline hydrolysis of the ester group in its precursor. The most remarkable property is the high solubility in a variety of solvents. The synthetic route allowed the full development of head-to-tail (HH-TT) linkages [177]. McCullough et al. reported the inclusion of –OH in bromine terminated HT-PHT [154]. The synthetic route involves the modification of the end group modified at the ω end through a cross-coupling reaction by using thienylzinc compounds. Therefore, THP hydroxy group is protected but can then be deprotected to obtain the –OH functionality at the ω e end. In another work, Lanzi et al. achieved the regioregular poly[3-(6-hydroxyhexyl)thiophene] (POH) through alkaline hydrolysis of the ester poly[3-(6-bromohexyl)thiophene] [153].

##### Reaction with Miscellaneous Groups 

Covalent functionalization with ferrocene redox groups on regioregular head-to-tail P3HT was achieved via substitution of –Br functionality in poly[3-(ω–bromohexyl)thiophene]. The methodology implied the dissolution of ferrocenecarboxylic acid in THF (N_2_ atmosphere) with the addition of 1,8-diazabicyclo[5.4.0]undec-ene at high temperatures (ca. 100 °C). A full spectroscopic characterization involving several techniques confirmed the functionalization of the P3HT-Br [178]. The inclusion of a tetraphenylporphyrin (TPP) in a poly[3-hexylthiophene-co-3-(6-bromohexyl)thiophene] (P(T6H-co-T6Br) copolymer (in different molar contents of 80:20 and 75:25) was reported by Salatelli et al. [179]. The reaction involved the replacement of bromine with TPP to obtain regioregular P(T6H-co-T6TPP) in DMF/THF in presence of dibenzo-18-crown-6. The final product has potential applications in solar cell devices due to the increase of the dye content without decreasing the solubility of the material compared to previous reports. PT multilayer films deposited on amine-terminated Si surfaces are also possible to generate due to the formation of secondary amine bonds on the surface amino (–NH_2_) by nucleophilic reaction [180]. The PT films should be functionalized with bromine groups. The covalent interaction can be confirmed using spectroscopic techniques (UV-Visible, XPS). Additionally, the layer-by-layer deposition can be monitored using atomic force microscopy. Esterification reactions between graphene oxide (GO) and the alkoxy poly[3-(2-(2-(2-(2-(diethanolamine)ethoxy)ethoxy)ethoxy)ethoxy)thiophene] (PD4ET) allowed the obtention of PD4ET-*g*-GO [181]. The synthetic approach consisted of the dispersion of GO in thionyl chloride, helped by ultrasonic and stirring, followed by the addition of the PT derivate and triethylamine under the same conditions. The most important difference from previous reports is the fact that in this work, effectively covalent interactions between the two components (PT and GO) were confirmed (by XPS, Raman, FTIR, XRD, among others).

The target property in PT functionalization is an improvement on the semiconducting and optoelectronic properties for applications on OLEDs, solar cells, etc. The attached groups could affect the conjugation length through the steric and/or inductive effect. Moreover, simple attached groups (e.g. –NH_2_) could be used to link more complex molecules (e.g., phthalocyanines) with optoelectronic properties of their own, which are combined with those of PT. Optoelectronic properties of PT improve by functionalization, specifically the chemical stability under illumination for the fluorine-containing polymers.

#### 2.2.3. Reactions with Substituted Poly(3,5-dioxythiophene) (PEDOT) 

The most widely studied substituted poly(thiophene) is poly(3,5-dioxythiophene) (PEDOT) [182]. The unmodified monomer unit contains only highly inert ether groups [183], making it a chemically stable polymer [184]. However, polymers made of dioxy rings with pendant reactive groups can be produced and used as platforms for covalent postmodification of PEDOT [185,186]. Specifically, “click” chemistry of active PEDOT chains [149,187,188,189,190,191] allows the production of conductive polymers with properties tuned to the applications in a simple manner. The functionalized polymers show tuned electronic [192], or electrochromic properties [193]. Wu et al. transformed a chloromethyl-EDOT into an alkene group and used the thiol-ene reaction to attach simple thiols (mercaptopropionic acid) or complex moieties (proteins, aptamers) [194]. The polymer films, modified with bioactive macromolecules, show enhanced biological activity. Finally, functionalizable derivatives of EDOT were polymerized in the solid state, and active PEDOT-based chains were produced [195]. However, to the best of our knowledge, the polymers were not used for postmodification. Bauerle and coworkers developed a versatile methodology for the postpolymerization of poly(azidomethyl-EDOT) films through reaction with terminal alkynes under mild heterogeneous conditions [149]. The approach overcomes limitations that appear when the substituents reacted with the monomer prior to polymerization (monomer modifications). The modification was carried out through click reaction of the PEDOT derivate in presence of the alkynes and Cu(CH_3_CN)_4_ PF_6_, using ACN, TFH, or benzonitrile depending on each alkyne. Successfully, the methodology allowed the inclusion of different functional moieties, including alkyl chains, different electron acceptors, and electron donors [188]. Other authors functionalized a copolymer film P(EDOT-N3-co-3T) with 1-hexyne and alkyne sulfonate via cycloaddition click reaction [191]. They used [Cu(NCMe)_4_]PF_6_ and copper (powder) as catalysts and DMSO as solvent. As a result of the PT derivate functionalization, tuning of the surface polarities and wettability properties of the films was achieved. A similar approach was also carried out for the functionalization of poly(3-(3,4-ethylenedioxythiophene)prop-1-yne) -poly(pyEDOT)- [196]. Additionally, starting from azide-containing thiophene obtained via SN_2_ substitution and their subsequent electropolymerization, a poly(thiophene)-azide can be synthesized. Starting from this modified PT, a glycosylated poly(thiophene)-lactose was obtained via click reaction with a disaccharide (lactose). Evidence of the success during the functionalization was obtained using XPS measurements and the contact angle technique, making the new modified CP substrate a promising tool for biointerface applications [197].

The target properties of PEDOT are the electrochromism and the use of the polymer as hole injection layer in OLEDs and solar cells. In the case of electrochomics, the attached functional groups affect the energy of the HOMO to LUMO transition. In that way, the color of the dark/clear states can be changed. The ability to tune the color allows the production of multicolor electrochromic pixels (red–green–blue) with the same base polymer (PEDOT). On the other hand, as PEDOT has absorption on the near infrared (NIR) range, smart windows which are transparent in the visible and switchable in the NIR allows to control the heating/cooling by radiation of houses or satellites. PEDOT is an excellent material for the hole injection layer [183]. Changing the absorption of that layer, it is possible to produce OLED pixels of more defined colors when the emitting layer has multiple bands. Moreover, both the electronic properties and the stability of the injection layer could be improved by functionalization. 

### 2.3. Polypyrrole Functionalization

The work on the postfunctionalization of PPy has been much more limited than that on PANI or PT. It is summarized in Table 3.

Camurlu et al. made clickable polypyrrole containing reactive azide groups [199]. The pyrrole monomer has a pendant alkyl chain terminated with an azide group (–N_3_). The monomer is electrochemically polymerized to give films. The azide groups react in a “click” fashion with alkynes to give cycles [207]. The polymer is reacted with ethinylferrocene to give films with clear electrochromism. The same authors also made a clickable poly(thiophene-pyrrole-thiophene) containing reactive azide groups [198]. First, they synthesized a monomer containing a central pyrrol linked with two thiophenes. The pyrrole unit has a pendant alkyl chain terminated with an azide group (–N_3_), which reacts in a “click” fashion with ethinylferrocene to give a film with clear electrochromism. 

Raicopol et al. used the well-known method of aryl radical formation through reduction of diazonium salts [208] to attach aromatics to PPy [200]. Given that reductive potentials were used, the PPy was in its aromatic form and a Gomberg reaction occurred [209]. The main factor in the effective functionalization is the reduction potential of the diazonium ion, which is related to the electronic effects of the group attached to the benzene ring. The method could be used to link reactive (e.g., –COOH) groups, which could then be used to bioconjugate biomolecules. Raicopol et al. used the same method to attach a p-nitroazoazobenzene moiety to PPy [206]. Since the azo group in p-nitroazobenzene can change from *trans* to *cis* through light excitation, the UV–visible spectrum of the functionalized PPy changes upon irradiation. Jang et al. sulfonated PPy via electrophilic aromatic substitution with chlorosulfonic acid [201]. After reaction, the chlorosulfonated PPy was hydrolyzed by heating it in water (100 °C for 4 h). Then, the sulfonated PPy was further doped with organic sulfonates (e.g., DBSA), which are known to help solubilize PPy in organic solvents [210]. Some blends (e.g., PPy-SO_3_^−^/DEHS) show up to 3% solubility in water, with good (0.2 S cm^−1^) conductivity and good solubility in organic solvents. 

Das et al. used the same procedure as Jang et al. [201] but did not heat the material in water and wash it with methanol, likely forming methylsulfones or maintaining the chlorosulfonate groups. The FTIR spectra could not distinguish between those groups. The material is stable, and it is used in DMFC devices. Gustafsson et al. showed that PPy reacts irreversibly with ammonia, changing the resistance and electronic spectra [211]. They suggest that this could be a nucleophilic addition of ammonia and/or of OH^−^ (produced from ammonia and residual water). Based on such information, Bieniarz et al. showed that strong nucleophiles are able to add to the oxidized form of PPy particles [203]. Therefore, thiols or amines are able to add nucleophilically to PPy, in a similar way to PANI. However, they failed to identify the reduced (and modified) form of PPy as the product of the addition. They suggested that by linking mercaptoacetic acid to the PPy, it would be possible to conjugate biomolecules via amide formation. It is noteworthy that PPy is the CP of choice for biomedical applications [212], likely because it maintains good conductivity at neutral pH (unlike PANI). However, biological fluids are full of thiols (cysteine, glutathione) and aminoacids with amino active amino groups (e.g., lysine), which could irreversibly react with PPy, changing its physical properties. Frontera et al. used similar reactions to combinatorially modify films of PPy (supported on PE) with different nucleophiles [204]. Reactions were observed in all the thiols used. The physical properties (conductivity, water contact angle) of the films modified by two nucleophiles together do not show values intermediate to those of the films modified by each nucleophile alone but are larger or smaller. Such synergic effects depart from the usual rules of combinatorial chemistry. 

Miodek et al. modified electrochemically polymerized films of PPy [205]. They subjected the PPy film to anodic potentials in the presence of a dendrimer (PAMAM G4). They detected the immobilization of the dendrimer, which was then modified (using bioconjugate peptide chemistry) with ferrocene and DNA strands. They assumed that the aliphatic amino groups of PAMAM form radical cations which attack the PPy. However, at the potentials involved (up to 1.1 V_RHE_), PPy is oxidized and could easily suffer nucleophilic addition by the amino groups. Moreover, radical cations of aliphatic amines are unstable (unlike aromatic ones) and easily produce carbocations [213]. 

The target property of PPy functionalization is its biocompatibility towards its application biomedicine. Therefore, biomolecules and related molecules (e.g., mercaptoacetic acid) were attached to the CP backbone. In a similar way, amino-terminated dendrimers are linked to PPy, likely by NCA. All surface modifications increase the biocompatibility and allow the linking of active biomolecules (e.g., enzymes). Since PPy is insoluble in aqueous solutions and common organic solvents, the improvement of solubility by functionalization is also an important goal. Surface reactions on nanoparticles of PPy which attach ionic groups (e.g., sulfonate) allow the stabilization of high-solid dispersions of nanoparticles towards producing conductive inks. Analogously, surface reactions on PPy films allow changing of the wettability and related interactions, modulating biological cell adhesion. For tissue growth, good cell adhesion is desirable, while the opposite is true when the film is the coating of a biomedical device for which pathogenic biofilm formation must be avoided. 

### 2.4. Functionalization of Other Conducting Polymers

Apart from the highly studied PANI, PT, and PPy, few studies of functionalization by covalent postmodification of other conducting polymers have been published. The data are summarized in Table 4.

#### 2.4.1. Poly(acetylene)

The first “modern” conducting polymer is poly(acetylene) (PA) [214]. While PA is highly reactive due to the conjugated C=C bonds, the addition reaction produces sp^3^ carbon in the chain. The extended conjugation is interrupted by those defects, drastically decreasing the conductivity. On the other hand, each monomer unit contains two H, which can hold substituent groups. Those groups can be then functionalized. A substituted PA (poly(phenylacetylene)-PPA-) was synthesized through polymerization of derivatives of 4-ethynylbenzoic acid [215]. Those polymers containing activated ester groups are then reacted with aromatic amines. A shift in the optical absorption bands, upon reaction with amines, suggests an effect on the conjugated backbone of the polymers. 

**Table 4 polymers-15-00205-t004:** Summary of reaction used in the postfunctionalization of other conducting polymers.

Conducting Polymer	Added Group	Reaction Kind	Reaction Form	Reactants	Target Property	Ref.
PA (PPA)	–CONHAr	Amidation	Bulk	Aromatic amines	Optical absorption	[212]
PP (P(p-dMeoBz))	–CN	Nu substitution	Film	CN^-^	Electrochemical	[213,215]
PPV	–HN–R–NH_2_	Nu substitution	Bulk	H_2_N–R–NH_2_	Cytocompatibility	[216]
PPV (MEH-PPV)	Multiple	(i) DCC catalyzed conjugation with –COOH (ii) “click” alkyne-azide	Bulk	various	Synthesis	[217]
PPV (MEH-PPV)	-Phtalocyanine	“click” alkyne-azide	Film	Functionalized phtalocyanine	Solar cells	[218]
PPV (DOH-PPV)	–X, Succinimide	Electrophilic addition	Bulk	NBS NCS	Fluorescence yield Solubility	[219]
PPV	–HN–R–NH_2_	Silane chemistry	Bulk	diamines	Gene therapy	[220]
PPV	–Si–O–Si–NR–NH_2_	Silane chemistry		Silane diamines	Cell adhesion	[221]
PPE	OEG-oligopeptide	Reactive group (–COOH)	Film	OEG + oligoppetide	Gene therapy	[222]
PFO	PEG block	Terminal group	Bulk	PEG	Fluorescence Emission	[223]
PFO	Monosaccharides	Thioether	Bulk	Monosaccharides	Biocompatibility	[224]
PFO	PEG	“click” Diels-Alder	Bulk	Transcyclooctene	Fluorescence	[225]
PFO	PEG	“click” Diels-Alder	Bulk	Transcyclooctene term. PEG	Fluorescence 3D crosslink	[226]
P(FO-alt-T)	–X (halogen)	NCA	EchemFilm	X^−^ (halide)	Optical	[226]

#### 2.4.2. Poly(phenylene)s

Fabre and Simonet electropolymerized (alone or as a copolymer with 3-methylthiophene) an activated (for SEAr) derivative of benzene (p-dimethoxybenzene) [217,218]. The cation radicals produced by anodic oxidation of the polymer chains suffer nucleophilic attack by CN^−^, which replaces one of the methoxy groups (–OCH_3_) to be removed as methoxide ion. The reaction is a nucleophilic substitution of the cation radicals of dimethoxybenzene, which also occurs in the monomer [219]. The replacement of the electron donating methoxide (–OCH3) by the electron-withdrawing cyano (–CN) group increases the potential for the formation of polarons in the polymer. The cyano group could be easily transformed into amine, carboxylic acid, or amide groups. 

#### 2.4.3. Poly(phenylenevinylene) (PPV)

Poly(phenylenevinylene) is a widely used semiconducting polymer [220]. Since it contains an aromatic ring in each monomer unit, it can be functionalized by reactions of activated groups linked to the aromatic ring. Since alkoxy derivatives of PPV (e.g., poly(2-methoxy-5-(2′-ethylhexyloxy)-1,4-phenylene vinylene, MEH-PPV) are widely studied as optoelectronic materials, the alkyl chain can be used as linker to reactive groups for functionalization. Duchateu et al. developed a general method of postfunctionalizing MEH-PPV copolymers [217]. A MEH-PPV monomer in which the methoxy group has been replaced by an alkoxy chain terminated in carboxylic groups was synthesized and copolymerized with MEH-PPV monomer. Then, using DCC catalyzed conjugation vinyl, propynylphenyl groups, methacrylate, and propargyl groups were attached. Additionally, MEH-PPV copolymers with ATRP and dithiocarbamate initiator groups were produced. The propargyl group was also used to further functionalization through “click” chemistry with an azide-functionalized reagent. The method was used to attach Zn phtalocyanines to MEH-PPV and broaden the optical absorption window of the base polymer [221]. On the other hand, Holdcroft et al. treated a related polymer (poly(*p*-2,5-dihexyloxy−phenyelenevinylene) (DHO−PPV)) with N-halosuccinimides (NXSuc) [222,227]. In addition to electrophilic substitution on the aromatic rings of PPV, halogen and succinimide (in CHCl_3_) add to the vinylene groups, forming sp^3^ defects. As expected, for high NXSuc/PPV ratios (>0.75), the solubility increased but the fluorescence yield decreased dramatically due to the loss of extended conjugation. However, for intermediate NXSuc/PPV ratios (0.5 to 0.75), the solubility is improved, and the fluorescence yield is enhanced (up to 150%) due to exciton confinement between defects and the reduction of intrachain quenching due to reduced chain aggregation. Another approach is to synthesize block copolymers of phenylenevinylene with vinylene moieties with pendant reactive groups (e.g., silane). The reactions can be used to prepare surface active oligomers [223] or to modify the surfaces of nanostructures [224]. Either the optoelectronic properties (e.g., luminescence), or the surface properties of films (e.g., wettability) or nano-objects (e.g., colloidal stability) could be tuned by polymer functionalization. 

#### 2.4.4. Poly(phenylene ethynylene)

Functionalized poly(phenylene ethynylene)s (PPEs) have been synthesized [225]. The –COOH groups are separated from the conductive chain by oligo(ethylene glycol) (OEG) spacers. A 14-mer peptide (Lys(DNP)-GPLGMRGLGGGGK) is attached to the spacer by peptide bioconjugate reactions. The peptide quenches the fluorescence of the PPE. The fluorescence is restored via treatment with trypsin which cut the oligopeptide, making the polymer a fluorimetric sensor for proteases. 

#### 2.4.5. Poly(fluorene) (PFO)

Poly(fluorene) (PFO) is a semiconducting polymer used in optoelectronics [226]. Marsitzky et al. synthesized polyfluorene chains terminated in alcohol [228]. The PFO block was then copolymerized with PEG to produce rod–coil block copolymers, which produced organized systems with high fluorescence emission. Xue et al. used a thioether approach to attach monosaccharides to PFO chains [229]. The resulting functionalized polymers showed large solubility in water and allowed the interaction of biological systems and semiconducting polymers. Kardelis et al. used “click” chemistry (inverse-electron-demand Diels–Alder) with trans-cyclooctene (TCO) to functionalize a CP related to poly(tetrazine-co-fluorene) (PFO) [230]. The polymer showed enhanced fluorescence intensity compared to poly(tetrazine). Moreover, linear chains and a 3D foam can be produced by grafting TCO-functionalized poly(ethylene glycol) chains. Finally, direct reactions on the fluorene units have been used to modify PFO nanoparticles to produce fluorometric sensors of explosive nitro compounds (e.g., picric acid) [231]. Inagi uses electrochemically driven NCA to functionalize an alternating copolymer of thiophene and fluorene [167], which reacts in the thiophene unit adding halide groups (e.g. –Br), likely by nucleophilic addition. The goal of the functionalization of PFO is improving its optoelectronic properties. Direct linkage of groups to the fluorine unit changes the optical properties (e.g., fluorescence emission) of the polymer.

## 3. Conclusions

Conducting polymer (CP) functionalization is an important field with a wide variety of technological applications under development. Among the common CPs, functionalized polyanilines and polythiophenes constitute most of the work. It is surprising that methods for functionalized polypyrroles (PPys) are somewhat limited given the important role of the polymer in biomedical applications, likely due to its good electroactivity/conductivity at neutral pH. In that sense, functionalized PPys should be easily linked to biomolecules by well-known bioconjugate chemical reactions. Work on other polymers (PA, PPV, PP, etc.) is even more limited. 

Direct reactions on the chain are usually used for the modification of polyanilines (PANIs). The main goal is to improve the solubility in organic solvents and aqueous solutions to be able to coat materials or cast films. In that sense, polyaniline sulfonated via electrophilic aromatic substitution while dissolved in concentrated acid is likely the most widely applied functionalized conducting polymer. The good electroactivity/conductivity in acid/neutral aqueous solution and nonaqueous electrolytes of functionalized PANI allow its use in different applications. Among them are uses as electrode material in electrochromic devices, batteries, bioelectrodes, interlayers in OLEDs, enzymatic electrochemical sensors, anticorrosion coatings, and photovoltaic cells. The high solubility in water and extended π conjugation allow use in the noncovalent functionalization of graphene and carbon nanotubes and as a stabilizer of their colloidal solutions. The strong acidic character of the sulfonate group makes it useful in polymer electrolyte membranes and acid catalyst in organic chemistry. Finally, its compatibility with other polymers is used to form composites with hydrogels and nanofiltration membranes. It is remarkable that while use as an electrode requires good conductivity (>0.01 S cm^−1^), other properties do not require it and are typical of polyelectrolytes. Sometimes, it seems that the qualification “conducting polymer” obscures the polymer nature of the material, which could give better properties. Another conclusion that can be drawn from the data reviewed is that synthetic methods applied to CP have not kept pace with the developments in modern organic chemistry. Most reactions used have been known since the 19^th^ century (e.g., S_E_Ar sulfonation). It should be remembered that the main chains of CPs contain aromatic (or heteroaromatic) rings, while most modern organic chemistry deals with aliphatic reactions. Accordingly, they use solvents (e.g., cc. H_2_SO_4_) and reactants (e.g., SO_3_), far from acceptable in modern (green) chemistry. Nor have novel experimental methods (microwaves, ultrasound, mechanochemistry) been used. However, in the few studies of reactions with CPs bearing reactive groups (e.g., ethynyl), modern reaction methods (e.g., “click” reactions) are used. In fact, a green alternative for the synthesis of sulfonated polyaniline exists—this is the nucleophilic addition, which could be made in water with nontoxic sulfite. However, it is seldom used to produce sulfonated polyaniline for technological applications. Moreover, unlike S_E_Ar sulfonation, NuA can be performed heterogeneously, allowing the modification of the surface of films or nano-objects without affecting the inner part of the polymer object. This is a capability that also set this method apart from the alternative method of producing functionalized PANIs, which is oxidative homopolymerization of previously functionalized monomers. It should be mentioned that in several cases, it has been shown that the properties (e.g., electronic conductivity) of equivalent polymer chains (e.g., poly(2-bromoaniline)) are quite different when produced using postfunctionalization (e.g., S_E_Ar bromination of PANI) rather than polymerization of the functionalized monomer. It seems that the real polymer chains are quite different in each case and that the postfunctionalized polymer is closer to the linear polymer chain. In the case of polythiophene, some direct reactions with the polymer chain were studied, but the main line of work implies functionalization of PT chains already bearing reactive groups (e.g., –COOR). Those polymers are produced via homopolymerization or copolymerization of already functionalized monomers. It is not clear why the polymerization of substituted anilines (ortho or meta) is more difficult than substituted (3-) thiophenes or pyrroles, but this is where the experimental evidence leads. The functionalization of such precursors uses all the breadth of modern organic chemistry. Its main goal is not solubility, which has already been solved with monomer functionalization (e.g., P3HT), but better electronic properties and interaction with the media. In that sense, electrochromic, semiconducting materials working both in “clean” conditions and inside biological systems have been developed using postfunctionalized PTs. In the case of PPy, only some of these methods have been applied. Postfunctionalization reactions seem to depend more on the presence of the reactive group than on the rest of the molecule attached. Therefore, parallel reactions with different reactants bearing the same reactive moiety can be easily performed in combinatorial fashion. The method has been applied only twice, first by coupling combinatorially synthesized azo dyes to PANI (one dye per polymer) and then by reacting multiple nucleophiles to the same PPy chain. Both methods show promise in the fast creation of functionalized polymer libraries. The synergic effect observed in the physical properties (conductivity, wettability) when multiple thiols were used to modify the same polymer remains to be explained. 

The model described in the introduction, which assumes that common reactions exist for different CPs, has not been fully studied. There are reactions (e.g., nucleophilic addition to oxidized CP chains) that are known to work in all materials. Others (e.g., SEAr, especially coupling with diazonium salts), despite only having been demonstrated fully for PANI, are known to work for pyrrole and thiophene monomers and should work on them. Finally, the powerful method of producing polymers with reactive groups, which has been extensively used in PT and related polymers, should be extended in full to PPy, for which polymerization of substituted monomers is a simple task. In the case of PANI, the difficult polymerization of functionalized anilines hindered the use of substituted monomers to produce reactive precursor polymers. However, poly(2-bromoaniline) has been used as precursor of phosphonated PANI. One problem common to polymer functionalization in general is the lack of characterization techniques that could identify the new bonds formed. In general, only attachment of the added group is verified, usually not even quantitatively. Since NMR is difficult to apply to solid (usually insoluble) polymers, other spectroscopies (FTIR, Raman) must be applied. Quantitative identification of functional groups can be made with XPS, but the technique is unable to detect the position of the attack, and even the band assignment is only based on known compounds. Even FTIR shows shifts due to neighboring group effects. The use of model compounds, which can be analyzed using NMR and FTIR, is an interesting approach that was used in the early work of Wrighton et al. [52] and seldom utilized in later studies [55]. The use of modern mass spectroscopy (e.g., MALDI-TOF) could be relevant to identify the new entities formed by functionalization. 

## 4. Patents

Barbero: C.A.: Miras, M.C., Salavagione, H.J. Polianilinas nitrosadas. Procesos de produccion, soluciones y peliculas, Argentine Patent (INPI), Nr. AR021531B1, 3 December 1999.

Barbero, C.A., Miras, M.C., Morales, G.M., Salavagione, H.J., Grumelli, D.E. Proceso para la sintesis de polianilinas modificadas por adicion nucleofilica, Argentine Patent (INPI) Nr. AR021531B1, 8 December 1999.

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
