# Peer review of "Functionalization of Conductive Polymers through Covalent Postmodification"

_polymers, 2022, doi:10.3390/polym15010205_

Round 1
Author Response
Answer TO REVIEWER #1
The manuscript: “Functionalization of conductive polymers by covalent post modification” by S.B. Abel et al., reports the review of conducting polymer post-functionalization for various applications. The authors focused mostly on: polyaniline (PANI), Polypyrrole (PPy) and polythiophene (PT). The authors discussed two approaches: i) direct functionalization and ii) reaction with substituted polymers bearing reactive groups attached to the chain. The later requires homo or copolymerization of substituted monomers, which is possible with pyrrole and thiophene, but difficult with aniline. Direct reaction could occur with the aromatic ring (electrophilic aromatic substitution, SEAr) of the more reduced forms. This kind of reaction used to functionalize PANI by sulfonation, halogenation and coupling with diazonium ions. In PANI and PPy, the hydrogen in the nitrogen can be electrophillically substituted or the N-H group could act as nucleophile giving nucleophilic substitution (SN) with activated alkyls or acyl groups. They could also give aromatic nucleophilic substitution (SNAr). The other direct reaction is the nucleophilic conjugate addition on the oxidized form of the polymer, which is the only reaction that confirmed experimentally to work for all the three CPs. In the case of PT, the main method has been indirect functionalization of chains by reaction with reactive groups present in the monomer. The authors discussed a wide range of issues related to the functionalization of a number of important polymers such as PANI, PP, and PT. It should be noted that the literature used in this review is very extensive. The results presented by the authors in this review may be of interest for research in the field of conducting polymers and polymer electronics. In my opinion, the topic and materials of this review are relevant. The review as a whole makes a good impression. At the same time, some questions remain about this article.
We really appreciate the comments of the Reviewer about our manuscript. His/her valuable suggestions have been taken into account in the current version with the aim of improving our article. We have addressed all the points raised by the Reviewer as is followed described.
- I found that the review is addressed mainly to chemists, not physicists. In my opinion, the connection between each type of functionalization of polymers and the improvement in the characteristics of device devices based on them should be emphasized.
Several publications detail only the synthetic method. However, when the publications reviewed measure and/or discuss improvements in properties (e.g., solubility, fluorescence, wettability, etc.), a paragraph was added at this stage in each subsection or section discussing the target properties and how the functionalization is able to improve the properties.
- It would be useful to add some data on the functionalization of such polymers, which are very important for organic electronics, such as PPV, PFO, and some others.
The reviewer is right. To be comprehensive, a whole section was added discussing the small number of publications describing post-polymerization functionalization of other conducting polymers (PA, PP, PPV, PPE, PFO).
- I also recommend adding a short summary at the end of each section to more clearly state the goals of the functionalization experiments described in the section.
A paragraph is added to the different subsections/sections describing the target properties and functionalization's effect on improving such properties.
- Finally, I recommend improving the style of this manuscript, especially in the Introduction, to avoid duplication (for example, on page 2, the double abbreviation FTIR, line 46). In my opinion, some reduction in the length of the manuscript may be useful.
The duplication mentioned above as well as others were solved. The manuscript was thoroughly checked and multiple corrections have been made, as can be observed in the file with “Track changes” visible. Regarding the reduction of the manuscript length, we shorten the abstract and delete several comments (including most footnotes) which are not essential. The conclusions were also shortened. However, to satisfy other suggestions, we have to add references and text.
In conclusion, I believe that the topic of this manuscript is consistent with the topic of Polymers. At the same time, the manuscript needs some revision. Authors should edit the manuscript in accordance with the guidelines mentioned above. The revised manuscript must be resubmitted for publication to Polymers.
Thanks again to the Reviewer for his/her appreciation. We edit the whole manuscript to asseverate the accordance with the journal guidelines. All the changes can be seen in “Track changes” in the current version.
Author Response
ANSWER TO REVIEWER #2
The article entitled “Functionalization of conductive polymers by covalent post modification” by Barbero et al accentuates the functionalization of conducting polymer using covalent post modification process. They discussed most of the conduction polymer functionalization. The collected review and discussion are enough to support their manuscript. The authors should carry out the following issues more systematically.
We would like thanks to the Reviewer for his/her valuable comments and suggestions with the aim to improve our manuscript for its publication. After a careful reading, we made substantial changes and editions in order to accomplish the standard and the guidelines of Polymers. All the changes can be seen in “Track changes” in the current version.
1- The title, abstract and content of discussion should more concise. It should easy to understand by readers perspective.
Due to the incorporation of text and references on the limited work on other polymers (PA, PP, PPV, PPE, PFO, etc.), we believe the title could be maintained without adding qualifications.
The abstract was shortened and completely rewritten
The discussion has been thoroughly checked and the sentences and footnotes not essential to the main subject removed.
2- The native English language correction should be carried out for readers understanding.
The final version of the manuscript was checked by a native scientist who make several corrections which can be observed in the current version with “Track changes” visible.
- Include uniform abbreviation form of PANI (ES,EA,PS,PA.,etc) and Tg in the abbreviation part?
Done. Besides, new abbreviations involuntarily omitted in the first version are now included.
- FTIR = Fourier Transform Infrared Spectroscopy is repeated twice in line number 47 and 48?
The Reviewer is right. The duplication was removed from the abbreviation list.
- Include the reference to support the discussion in line number 83, 100,101,121, 128.,191,198,570 and 667.
83, done and rephrased
100, done
101, done
121, it has already a reference [old 22]
128, an explanatory note is added
191, done
198, done
570, done
and 667, done
- Check the typo error in line number 138 (coions), 306 (=), 482 (*), 523 ( )), 530 ( subscript and superscript), 561 ( α and β), 565 (pirrolydine), 609 (nucleohilic), 695 (VSSCE.) , 710 (chorine), 713 (F->Cl->Br->I-), 714 (VNHE), 730 (CL-), 730 (Fe+3 as Fe3+), 781 (moetiy), 1000 (aueous.), 1005 (caracter), 1009 (thechnological), 1001 (electrlytesd), 1040 (all), 1047 (paralell), 1058 (Finallly), 138 (coions), reference 55 “pyrrolidin io” and reference 61 “Lowl” in table 1, Check the reference 54 temperature symbol in table 1,Make uniformity SO3-2 or SO32- in reference number 66 in table 1, Check the reference 71 “sinteshys” in table 1,Check the reference 175 and 178 “(-SO3-† and addition‡ )” in table 4.
All the aforementioned typo errors were corrected in the manuscript. The footnotes were deleted as they are not accepted by the journal.
- Full stop error in line number 172.
This full stop and others similar to it were solved.
- Recheck the sentence from line number 213-215.
Done.
- Include the formula for “sulfonitric” mixture in 293.
The composition of the sulfonitric mixture was clarified.
- Degree symbols should be inserted in reference number 64,65,67,68 and 69 in table 1.
Thanks for the observation. All the symbols were correctly written.
- In line 370,570,615,686,712,729, 963 the reference style is not matched with polymer journal.
The Reviewer is right. Probably, it was wrong due to the conversion of the file. Now, the reference style was modified in order to accomplish this with Polymers.
- Line number 400 write the chemical formula of bromide.
Done.
- In scheme 7 the structure of bipolaron is incorrect the structure should be corrected.
Corrected.
- Use uniform word in all the tables (Eg. variable or var) for readers better understanding.
We made the changes to make the table entries uniform.
- Line number 691-693 not clear and hard to understand. The sentence should be restructured for readers better understanding.
Corrected.
- Include correct chemical name of (3,4-diamino-2,2:5,2-polythiophene).
Done.
- Avoid using unnecessary close bracket and open bracket in every sentence it’s hard to read.
We modify the whole manuscript accordingly.
- Use some latest reference should be used and discussed for all polymers functionalization.
We have discussed all references, but in some sections we found that no new developments have been published recently.
- Include uniform conductivity unit all over manuscript.
The conductivity units were changed in agreement with SI uniformly throughout the manuscript.
- Should be used proper subscript and superscript for all the chemical formulas and ionic charges (oxidation state), hour, temperature and make uniformity.
All formulas were checked and corrected when necessary.
- The continuous scheme number or table number should be used: Instead of using see above and see below.
Done. Most of the positional references were not essential and were deleted.
- This review article should be thoroughly revised (all the spelling mistake, space bar error and make sentence clear to understand by the readers).
The manuscript has been now thoroughly checked by software and by a native speaker to avoid typos and grammar errors.
- Reference style should be adhered with polymer journals. ( DOI number are missing)
We modified the reference style in accordance with the guidelines of the journal, including the DOI numbers involuntary avoided in the first version of the manuscript.
Round 2
Reviewer 2 Report
Authors carried out all the comments raised by the reviewer. so the present form of revised Manuscript may be consider for publication